# Structure Tensor-Based Infrared Small Target Detection Method for a Double Linear Array Detector

**Jinyan Gao [1,*]**, **Luyuan Wang [1]**, **Jiyang Yu [1] and Zhongshi Pan [2]**

1   Institute of Spacecraft System Engineering, China Academy of Space Technology, Beijing 100094, China
2   Institute of Remote Sensing Satellite, China Academy of Space Technology, Beijing 100094, China
*   Correspondence: gaojinyan15@nudt.edu.cn

**Abstract:** The paper focuses on the mathematical modeling of a new double linear array detector. The special feature of the detector is that image pairs can be generated at short intervals in one scan. After registration and removal of dynamic cloud edges in each image, the image differentiation-based change detection method in the temporal domain is proposed to combine with the structure tensor edge suppression method in the spatial domain. Finally, experiments are conducted, and our results are compared with theoretic analyses. It is found that a high signal-to-clutter ratio (SCR) of camera input is required to obtain an acceptable detection rate and false alarm rate in real scenes. Experimental results also show that the proposed cloud edge removal solution can be used to successfully detect targets with a very low false alarm rate and an acceptable detection rate.

**Keywords:** small moving target detection; double linear array detector; cloud edge removal; structure tensor

## 1. Introduction

Dim small target detection is a major problem in numerous fields, such as infrared search and track (IRST) systems and external intrusion warnings [1–3]. Since the imaging distance is long in these applications, the target usually occupies only one or a few pixels [4–6], and there is insufficient texture and shape information for target detection [7–9]. Furthermore, the intensity value of the infrared target is very low due to reflection, refraction, the sensor's aperture diffraction effects and geometric aberrations [10–13]. Therefore, it is difficult to separate infrared small targets from complex backgrounds.

### 1.1. Related Works

Existing infrared target detection approaches can be divided into spatial, temporal, and spatio-temporal detection methods. Most spatial detection methods use spatial filtering techniques, and they are usually based on the assumption that the target has a larger intensity value than the background. However, this assumption does not always hold in real scenes [14–17]. The temporal detection methods usually use the temporal profiles of each pixel in a sequence of infrared images to extract the small target of interest. They have a good detection performance when a small target appears in slowly evolving backgrounds [18–21]. However, these methods often consume more time than single-frame detection methods. The spatio-temporal detection methods are complementary to the singular spatial or temporal detection methods [22–24]. They use features in both spatial (e.g., the gray difference feature) and temporal (e.g., the motion difference feature) domains to completely separate targets from clutter.

Several optical systems have been proposed to detect small infrared small targets over the past few decades [25–28]. They can be divided into two classes: scanning camera-based optical systems and staring camera-based optical systems. Scanning cameras have a relatively wide field of view and are suitable for early warning for large areas [29]. However, imaging in this way has a high time delay integration (TDI) in adjacent frames.

Consequently, it is difficult to perform data association, and the time for target discovery is long. On the other hand, the staring camera is usually used for target tracking as its imaging size is usually small [30], which makes it not suitable for searching in early warning.

### 1.2. Contributions

In order to overcome these limitations of traditional optical systems, we propose to use a double linear array detector to detect targets with cross-pixel moving. For a double linear array detector, two images (an image pair) are generated when the detector scans from the top to the bottom only once. Figure 1a shows an IR image pair acquired by a double linear array detector with a slowly changing cloud clutter. It not only reserves the wide field of view of traditional scanning systems but also reserves the short-time intervals in adjacent frames of staring systems. A double linear array detector has the following three advantages. First, it reserves the wide field of view of traditional scanning systems. Second, its exposure time in each pixel is longer than traditional scanning systems. Third, the interval between adjacent frames is shorter than traditional scanning systems, which makes it easy to perform data association subsequently estimate the velocities and relative positions of cross-pixel moving targets.

Our other task is to automatically detect targets with cross-pixel moving in image pairs acquired by a double linear array detector. Considering the special spatial arrangement and imaging modes of the detector (as analyzed in Section 2), the image pair is almost observed from the same solar angle and atmospheric conditions, and the slow change in cloud background can be almost negligible at a short interval. Therefore, the image differentiation-based change detection method is suitable for the detection of targets after image registration. Figure 1b shows the detection results after image differentiation. We can see the positive gray-scale value and the negative gray-scale value of a candidate target produced from an image pair in the filtered result, namely, positive and negative target pairs in our paper. The mathematical model of image differentiation is:

$$Dx_{ij} = x_{ij}(t_2) - x_{ij}(t_1) \tag{1}$$

where $x(t_1)$ represents the image acquired by the first linear array and $x(t_2)$ represents the image acquired by the second linear array. $i$ and $j$ indicates the location of the pixel. $Dx_{ij}$ represents the residuals after image differentiation. If the grayscale value of the candidate target in $x(t_1)$ is positive, the grayscale value of the candidate target in $x(t_2)$ is negative after image differentiation. We call them positive and negative target pairs in this paper.

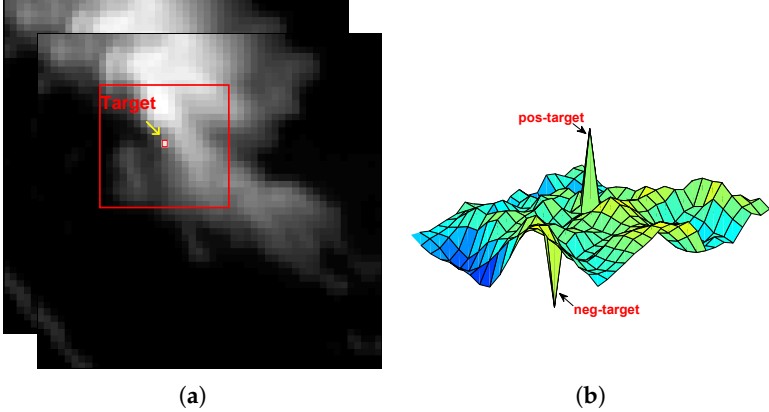

|   |   |
|:-:|:-:|
| (**a**) | (**b**) |

**Figure 1.** An example of image pair and positive and negative target pairs. (**a**) An image pair with slowly changing cloud clutter. (**b**) Local detection results after image differentiation.

Since false alarms after image differentiation are mainly caused by cloud edges [31,32], edge suppression has been found to be a useful spatial method complementary to our temporal change detection method [33–35]. The structure tensor has been widely used

for cloud edge suppression in recent years. Dai et al. [23,24,36,37] allocated the structure tensor as an adaptive weighting parameter to suppress strong cloud edges. Liu et al. [38] introduced the gradient direction diversity (GDD) method to suppress sharp cloud edges. The GDD measure is also inspired by the structure tensor. Li et al. [39] used the local steering kernel to encode the infrared image patch, as it can represent different intrinsic structures in different image regions (e.g., the cloud edge region, the flat region, the textural clutter region and the small target region). Thus, the structure tensor is used for cloud edge removal in our paper.

The overview of our proposed method is as follows.

- The structure tensor is used to detect infrared small targets, which is used as an adaptive weighting parameter to suppress strong cloud edges.
- Considering that using information of image sequences requires more prior information and a large amount of data processing, the temporal image differentiation filter is used to extract target pairs using movement information of the target.
- Adaptive thresholding-based constant false alarm (H-CFAR) is performed to obtain candidate targets, and data association is performed to extract positive and negative target pairs.

The rest of this paper is organized as follows. Section 2 analyzes the optical path and mathematical model of the double linear array. Section 3 presents the target detection model in detail. Section 4 tests the performance of our proposed method. The paper is concluded in Section 5.

## 2. The Double Linear Array Detector

For a double linear array detector, the region of interest in the instantaneous field of view ($IFOV$) can be measured twice in the same period of time once the detector scans from the top to the bottom. The optical path of a double linear array detector is shown in Figure 2. The device consists of a scanning mechanism, an optical system and a focal plane. The scanning mechanism mainly includes a pendulum mirror and a driving shaft, as wel as the focal plane constructed with two linear arrays arranged in parallel, as shown in the right part of Figure 2. Incident light containing the radiation energy information of targets and the background is first projected to the pendulum mirror; the driving shaft is then used to rotate the pendulum mirror. After several reflections and refractions in the optical system, the light finally converges to the focal plane to generate an image pair of the scene.

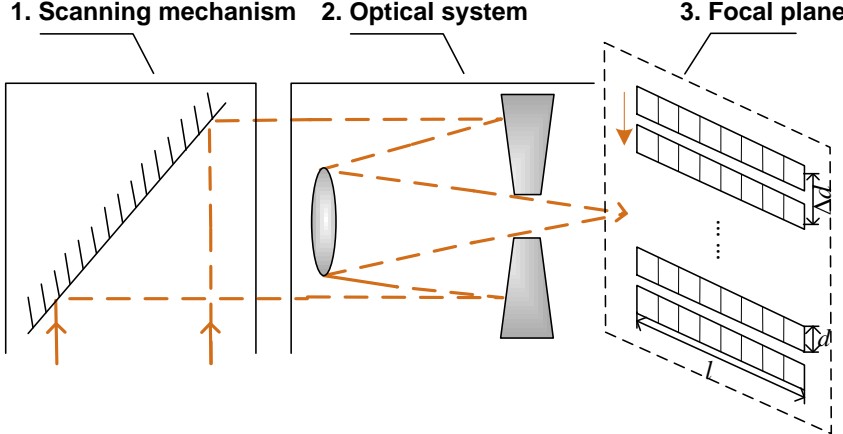

**Figure 2.** The optical path of a double linear array detector.

For a scanning system, the ground sample distance ($GSD$) determines the maximum spatial resolution of a camera and the minimum detectable velocity of a target. In addition, $GSD$ mainly depends on the instantaneous field of view ($IFOV$) in practice. The $IFOV$ is the angular cone of visibility of the camera and determines the area on the Earth's surface

that can be seen from a given altitude at a particular moment [40]. The geometry of the detecting system, including $GSD$, $IFOV$, camera height ($H$), pixel size ($d$), and the focus of the optical system ($f$) is shown in Figure 3, and the relationship between them can be expressed as:

$$GSD = 2H\tan(\frac{IFOV}{2}) = \frac{Hd}{f} \tag{2}$$

Considering that our detection method is based on two frames, and it has to use the distance constraint to associate the positive and negative target pairs, the following data association constraints are derived.

The target velocity in the image plane $v_{pixel}$ is used to predict the target velocity $v$ in real scenes. Since the dual linear array can only detect cross-pixel moving targets, the detectable velocity in the focal plane is:

$$\frac{d}{\Delta t} < v_{pixel} < \frac{l}{\Delta t} \tag{3}$$

where $d$ is the pixel size, $l$ is the width of the linear array, and $\Delta t$ is the time interval of linear arrays.

Then, the range of the detectable target velocity in real scenes is:

$$\frac{GSD}{\Delta t} < v < \frac{l}{\Delta t} \cdot \frac{GSD}{d} \tag{4}$$

The left and right sides of Equation (4) are the minimum and maximum detectable velocity of the target in real scenes, which are denoted by $v_{min}$ and $v_{max}$, respectively.

From Equations (3) and (4), we can obtain the distance constraints as follows:

$$\frac{v_{min} \cdot \Delta t}{GSD} < \Delta D < \frac{v_{max} \cdot \Delta t}{GSD} \tag{5}$$

where $\Delta D$ is the target moving distance in linear arrays.

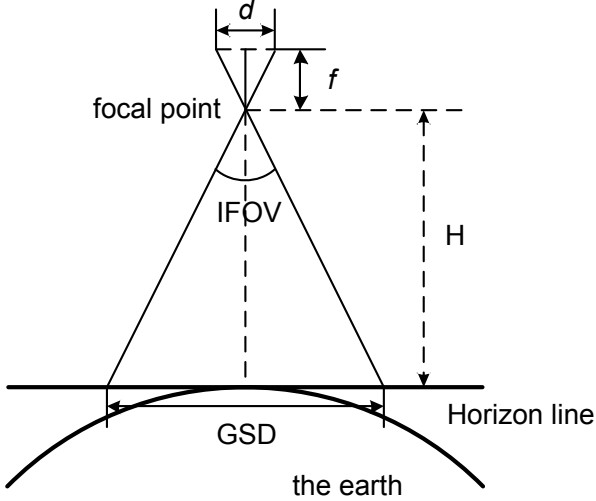

**Figure 3.** Geometry of the detecting system.

## 3. Target Detection Model

The proposed small target detection method for a double linear array detector is shown in Figure 4. We first use the structure tensor method to suppress cloud edges. Then, the temporal image differentiation filter is used to extract target pairs using motion information of the target. After background suppression, adaptive thresholding-based constant false alarm (H-CFAR) is performed to obtain candidate targets. Finally, data

association is performed to extract positive and negative target pairs using the constraints given in Equation (5).

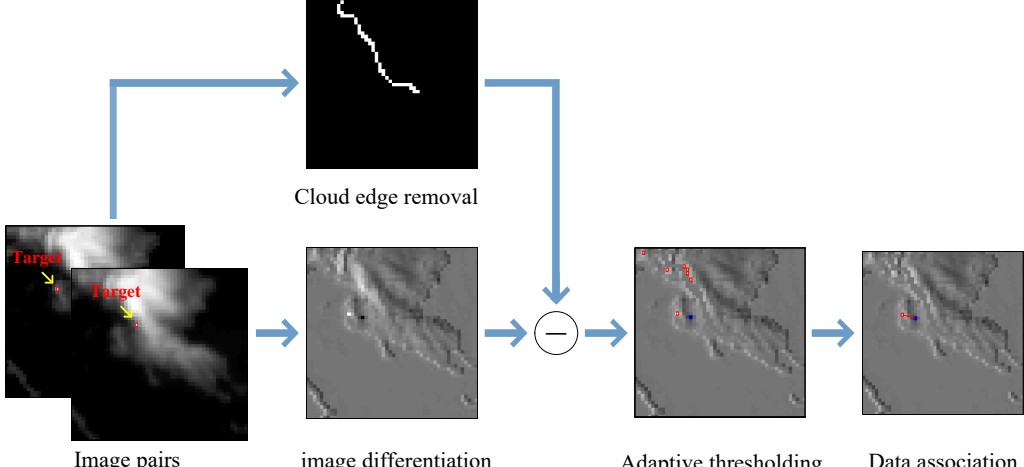

**Figure 4.** Overall flowchart for a double linear array detector.

The structure tensor is proposed based on the edge shock filter and variational functionals [41–43]. It turns out to be very effective for the enhancement of corner structures and presents different characteristics on homogeneous regions, edges, and texture regions of an image [44,45]. Therefore, it is used in our infrared small target detection method for cloud edge suppression in a single image.

The structure tensor is essentially a steering matrix [46,47]. It describes the local structural information about the image, and it can be represented as:

$$C_i = \sum_{\mathbf{x}_i \in \Omega_i} \begin{bmatrix} \frac{\partial I^2}{\partial x_{i1}} & \frac{\partial I}{\partial x_1}\frac{\partial I}{\partial x_{i2}} \\ \frac{\partial I}{\partial x_{i1}}\frac{\partial I}{\partial x_{i2}} & \frac{\partial I^2}{\partial x_{i2}} \end{bmatrix} \tag{6}$$

where $I$ represents the image and $\mathbf{x}_i = (x_{i1}, x_{i2})$ represents the two-dimensional coordinate vector of the central pixel in a rectangular window $\Omega_i$.

The steering matrix captures the principal directions of local texture from the gradient distribution in a small neighborhood (mostly $5 \times 5$ [48]). Therefore, the structure tensor $C_i$ can be first calculated by $G_i^T G_i$ with:

$$G_i = \begin{bmatrix} Z_{x_1}(x_1) & Z_{x_2}(x_1) \\ \vdots & \vdots \\ Z_{x_1}(x_P) & Z_{x_2}(x_P) \end{bmatrix} \tag{7}$$

where $Z_{x_1}(\cdot)$ and $Z_{x_2}(\cdot)$ denote the first derivatives along the horizontal and vertical axes, respectively, and $P$ is the number of pixels in the local window $\Omega_i$. However, since it is difficult to calculate the gradient distribution, the covariance matrix $C_i$ can be estimated by singular value decomposition [22,49] as:

$$C_i = \gamma_i U_{\theta_i} \Lambda_i U_{\theta_i}^T \tag{8}$$

where $\gamma_i$ is the scaling parameter. It is large in homogeneous regions but small in textured regions. $\theta_i$ is the rotation parameter; it defines the dominant orientation angle, $U_{\theta_i}$, as a rotation matrix. $\Lambda_i$ is the elongation matrix.

$$\gamma_i = \left( \frac{s_1 s_2 + \lambda''}{M} \right)^{\frac{1}{2}} \tag{9}$$

$$U_{\theta_i} = \begin{bmatrix} \cos\theta_i & \sin\theta_i \\ -\sin\theta_i & \cos\theta_i \end{bmatrix} \tag{10}$$

$$\Lambda_i = \begin{bmatrix} \sigma_i & 0 \\ 0 & \sigma_i^{-1} \end{bmatrix} \tag{11}$$

$$\sigma_i = \frac{s_1 + \lambda'}{s_2 + \lambda'} \tag{12}$$

where $\sigma_i$ is the elongation parameter. $\lambda' = 1; \lambda'' = 10^{-1}$. The structure tensor can be calculated using Equations (9)–(12).

The eigenvalues of singular value decomposition is denoted as $\lambda_1$ and $\lambda_2$. They can be used as two features to describe the local structural information [50]. The larger $\lambda_1$ is then $\lambda_2$; the measurement region is more likely a cloud edge region. Therefore, the cloud edge suppression measure can be defined as follows:

$$K = \exp(-h \cdot (\lambda_1 - \lambda_2)) \tag{13}$$

Figure 5 shows the cloud edge suppression results on images with three different shapes of clouds. It demonstrates that the proposed structure tensor-based measure achieves good performance in cloud edge suppression.

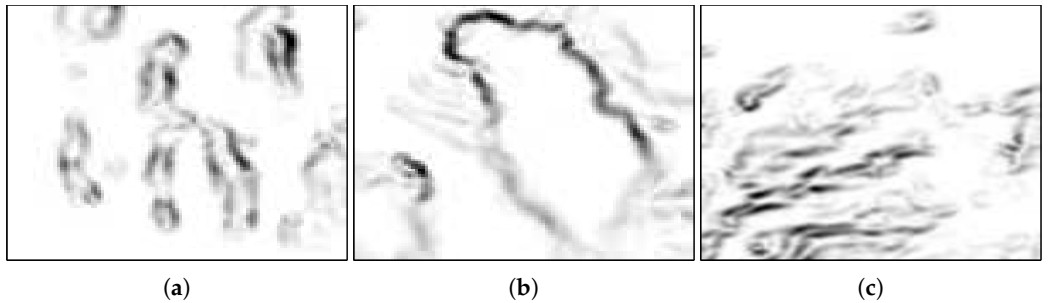

|   (a)   |   (b)   |   (c)   |

**Figure 5.** Illustrations of cloud edge suppression based on structure tensor measures. (**a**) Ragged cloud edge; (**b**) strong cloud edge; (**c**) fluffy cloud edge.

*Temporal Differentiation of Image Pairs*

Assuming that the noise in image pairs follows a Gaussian distribution $N(0, \sigma^2)$ and the registration error is zero, then the residual after image differentiation follows a Gaussian distribution $N(0, 2\sigma^2)$. Since the grayscale value of the residual image can be positive or negative, we use a bilateral filter to extract candidate points, the positive threshold $th\_p$ is used to extract positive candidate points, and the negative threshold $th\_n$ is used to extract negative candidate points. If both the positive and negative grayscale values of the target exceed the defined thresholds, the data association step is followed (according to Equation (5)). Finally, the detection rate $p_d$ and false detection rate $p_f$ can be calculated by Equations (14) and (15).

$$\begin{cases} p_d = p_d^+ \cdot p_d^- \\ p_d^+ = Q\left( \frac{th\_p - (T_{max} - \mu_{BG})}{\sqrt{2}\sigma_{BG}} \right) \\ p_d^- = 1 - Q\left( \frac{th\_n - (\mu_{BG} - T_{max})}{\sqrt{2}\sigma_{BG}} \right) \end{cases} \tag{14}$$

where $p_d^+$ and $p_d^-$ are the target detection rate with a positive gray value and the target detection rate with a negative gray value, respectively. $T_{max}$ is the highest intensity value in the target region, and $\mu_{BG}$ and $\sigma_{BG}$ are the average and the standard deviation of intensity

values in a residual image. $Q(x)$ is a right-tailed distribution function, as defined in Equation (16).

$$\begin{cases} p_f = N \cdot p_f^+ \cdot p_f^- \\ p_f^+ = Q\left(\frac{th\_p}{\sqrt{2}\sigma_{BG}}\right) \\ p_f^- = 1 - Q\left(\frac{th\_n}{\sqrt{2}\sigma_{BG}}\right) \end{cases} \tag{15}$$

where $p_f^+$ and $p_f^-$ are the target false alarm rate with a positive gray value and the target false alarm rate with a negative gray value, respectively. $N$ is the number of pixels that are possibly associated.

$$Q(x) = \int_x^{+\infty} \frac{1}{\sqrt{2\pi\sigma_{BG}^2}} exp\left\{-\frac{(\xi - \mu_{BG})^2}{2\sigma_{BG}^2}\right\} d\xi \tag{16}$$

To ensure that both positive and negative targets can be detected, the signal-to-clutter ratio (*SCR*) is defined as:

$$SCR = \frac{|T_{max} - \mu_{BG}|}{\sigma_{BG}} \tag{17}$$

Combining Equations (14)–(17), the relationship between the detection rate, false rate and *SCR* can be expressed as:

$$Q^{-1}\left(\left(p_f/N\right)^{\frac{1}{2}}\right) - Q^{-1}(p_d^{\frac{1}{2}}) = \frac{SCR}{\sqrt{2}} \tag{18}$$

The theoretic receiver operating characteristic (ROC) curves validated through Monto Carlo simulations for the double linear array detector are analyzed as shown in Figure 6. It can be seen that the detection probability becomes higher as the SNR increases. Specifically, When the SNR is 6.1 and the false alarm rate is $1 \times 10^{-4}$, our double linear array detector can achieve a detection rate of 97%; when the SNR is above 6.1 and the false alarm rate is $1 \times 10^{-5}$, our double linear array detector can achieve a detection rate of 93%.

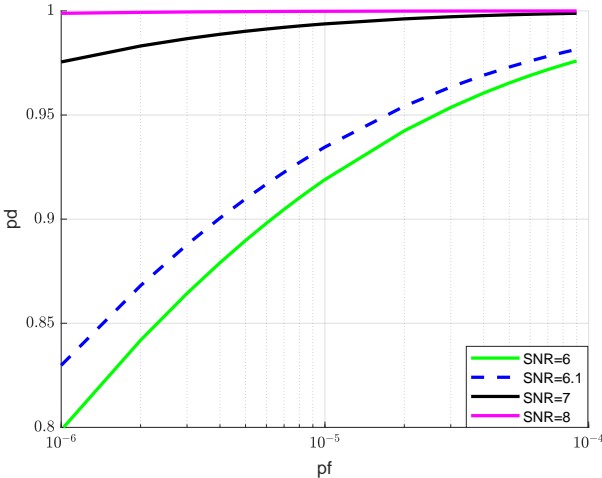

**Figure 6.** The theoretic ROC curves for the double linear array detector.

## 4. Experimental Results and Discussions

### 4.1. Simulation Scenes

In this section, experiments are conducted to test the performance of our method. Given an image, another image was generated with one pixel jittering in an arbitrary direction to simulate the image pair produced by the double linear array detector. Then, simulated targets were added into each image, the target position and motion direction

were randomly determined, the intensity of the targets was determined according to a specific *SCR*, and the target velocity was set to 2∼3 km/s. Next, the camera resolution was set to 1 km × 1 km, and the interval of an image pair was set to 2 s. Finally, considering that the target can move along the diagonal direction and the horizontal direction, the distance constraint for a target pair after clutter suppression is set to 2∼7 pixels, as shown in Figure 7.

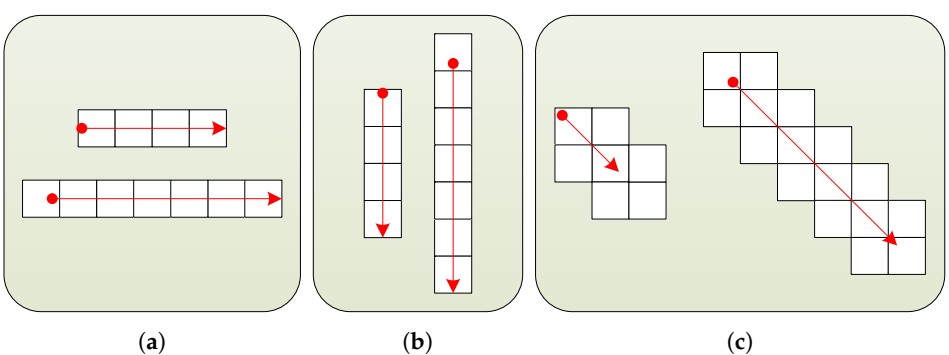

| (a) | (b) | (c) |

**Figure 7.** Schematic diagram of distance constraint for a target pair. (**a**) Horizontal direction, (**b**) Vertical direction, (**c**) Diagonal direction.

### 4.2. Experiments on Simulation Scenes

Three experiments were conducted on scenes with different shapes of cloud (i.e., Figure 8a–d, ragged cloud scenes; Figure 8e–h, strong cloud scenes; Figure 8i–l, fluffy cloud scenes), as shown in the first column of Figure 8. The second column in Figure 8 shows the filtered results after image differentiation, and the third column shows the filtered results after cloud edge suppression and image differentiation. It can be seen from the third column of Figure 8 that false alarms in dynamic cloud edges have been significantly reduced. Finally, accurate association results are obtained, as shown in the fourth column of Figure 8.

### 4.3. ROC Curves Evaluation

The ROC curves obtained through Monto Carlo simulations were used to test our proposed spatio-temporal method for a double linear array detector. An ROC curve represents the probability of detection $P_d$ as a function of the false alarm rate $P_f$. $P_d$ and $P_f$ can be calculated as:

$$P_d = \frac{n_t}{n_c} \tag{19}$$

$$P_f = \frac{n_f}{n} \tag{20}$$

where $n_t$, $n_c$, $n_f$ and $n$ represent the number of detected true pixels, ground-truth target pixels, false alarm pixels and the total number of image pixels, respectively.

As shown in Figure 9, at the cost of an increase in the false alarm rate, the detection probability is increased under a certain *SCR*. When *SCR* is increased, the detection probability is increased under a certain false alarm rate. It can be observed that the detection rate and the false alarm rate are a pair of contradictory variables. Furthermore, the theoretical ROC results are better than the results achieved by other two methods in real scenes. That is because the residual after background suppression does not completely follow a Gaussian distribution.

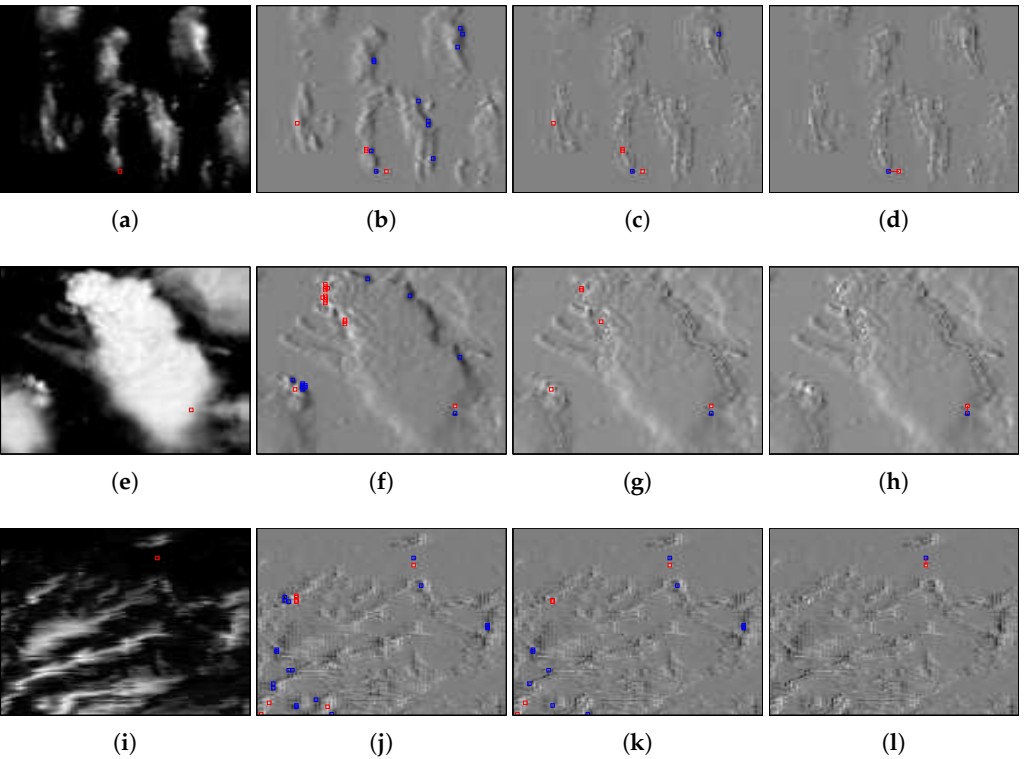

**Figure 8.** An example of results achieved by the proposed small target detection methods on three images. The red and blue boxes in each image represent the candidate points passing the positive and negative thresholds (as mentioned in Section 3), respectively. (**a**–**d**) Ragged clouds; (**e**–**h**) strong clouds; (**i**–**l**) fluffy clouds.

Table 1 shows the probability of detection in three experiments under different $SCRs$ and $P_f$. Table 2 shows the false alarm rates in three experiments under different $SCRs$ and $P_d$. We can see that with the cloud edge suppression method, the detection rate is significantly improved, and the false alarm rate is largely reduced.

**Table 1.** $P_d$ achieved by three methods under different $SCRs$ and $P_f$.

| SCR | $P_f = 5 \times 10^{-6}$ | | | $P_f = 10^{-5}$ | | |
|---|---|---|---|---|---|---|
| | Temporal | Spatial-Temporal | Theory | Temporal | Spatial-Temporal | Theory |
| 6 | 11.39% | 54.40% | 88.98% | 44.03% | 63.93% | 91.90% |
| 7 | 65.53% | 74.16% | 99.02% | 70.67% | 81.06% | 99.38% |
| 8 | 84.93% | 86.53% | 99.97% | 88.39% | 92.74% | 99.98% |

'Temporal' stands for the temporal image differentiation method; 'Spatial-temporal' stands for the method combining the spatial cloud edge suppression and the temporal image differentiation method.

**Table 2.** $P_f$ achieved by three methods under different $SCRs$ and $P_d$.

| SCR | $P_d = 85\%$ | | | $P_d = 90\%$ | | |
|---|---|---|---|---|---|---|
| | Temporal | Spatial-Temporal | Theory | Temporal | Spatial-Temporal | Theory |
| 6 | $3.30 \times 10^{-4}$ | $2.32 \times 10^{-4}$ | $3.00 \times 10^{-5}$ | $8.82 \times 10^{-4}$ | $5.14 \times 10^{-4}$ | $6.00 \times 10^{-6}$ |
| 7 | $4.34 \times 10^{-5}$ | $2.10 \times 10^{-5}$ | $2.00 \times 10^{-7}$ | $6.88 \times 10^{-4}$ | $6.16 \times 10^{-5}$ | $4.00 \times 10^{-8}$ |
| 8 | $4.97 \times 10^{-6}$ | $4.28 \times 10^{-6}$ | $8.00 \times 10^{-10}$ | $1.40 \times 10^{-5}$ | $8.08 \times 10^{-6}$ | $5.00 \times 10^{-9}$ |

'Temporal' stands for the temporal image differentiation method; 'Spatial-temporal' stands for the method combining the spatial cloud edge suppression and temporal image differentiation method.

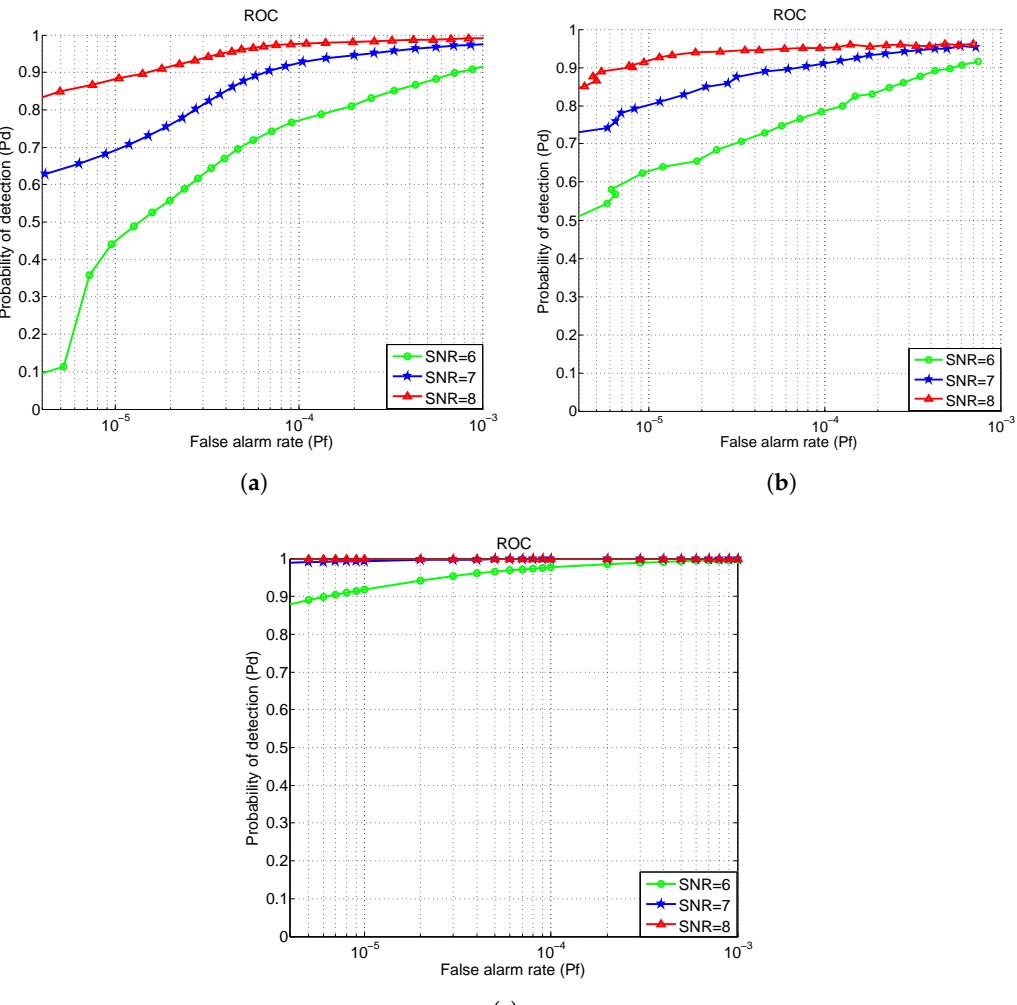

**Figure 9.** ROC curves achieved by three methods under different SCRs. (**a**) Performance achieved by temporal image differentiation. (**b**) Performance achieved by spatial cloud edge suppression and temporal image differentiation. (**c**) Performance achieved by theoretic analysis.

### 4.4. Evaluation of the Proposed Detection Framework

The data association results for positive and negative targets under different operating conditions are shown in Figure 10. From Figure 10a–p, we can see that after image differentiation and cloud edge suppression, the target pairs can be correctly associated, and no false detection is found in the association results. From Figure 10q–t, we find that one target can associate with two or more targets under a few conditions. To address this problem, we have to extend the dual linear array to a multi-linear array in future, and the false association can be removed by the direction of the target motion trajectory. Furthermore, the input SCR of the camera can be improved to handle this problem.

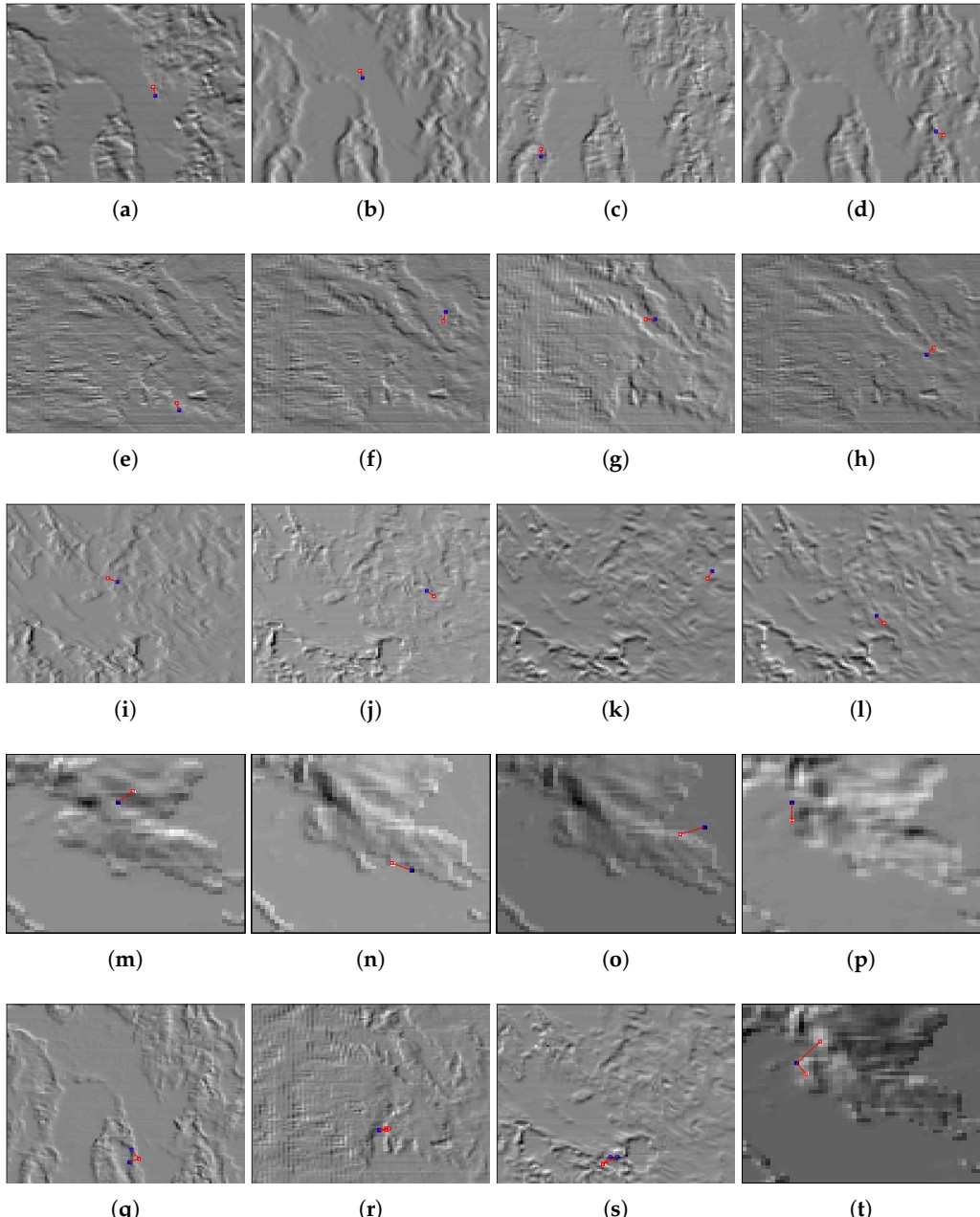

**Figure 10.** Thedata association results with targets in different positions in various complex backgrounds. (**a**–**p**) target pairs were associated correctly, (**q**–**t**) one target was associated with two or more targets.

## 5. Conclusions

This paper has presented a complete framework for small infrared target detection using a double linear array detector. First, a new double linear array detector was modeled to generate image pairs at short intervals. Second, considering the limitations of singular spatial or temporal detection methods, an image differentiation-based change detection method in the temporal domain was proposed combined with the structure tensor edge suppression method in the spatial domain. The experimental results showed that targets can be extracted accurately with a very low false alarm rate and an acceptable detection rate.

**Author Contributions:** Conceptualization, J.G. and Z.P.; methodology, J.G.; software, J.G.; validation, J.G. and J.Y.; formal analysis, J.G.; investigation, J.G.; resources, L.W.; data curation, Z.P.; writing— original draft preparation, J.G.; writing—review and editing, J.G.; visualization, J.G.; supervision, J.G.; project administration, L.W. All authors have read and agreed to the published version of the manuscript.

**Funding:** This research received no external funding.

**Data Availability Statement:** Not applicable.

**Conflicts of Interest:** The authors declare no conflict of interest.

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
