# Peer review of "Structure Tensor-Based Infrared Small Target Detection Method for a Double Linear Array Detector"

_remotesensing, doi:10.3390/rs14194785_

Round 1

Reviewer 1 Report

Title: Structure tensor based infrared small target detection method for a double linear array detector

Authors: Jinyan Gao, Luyuan Wang, Jiyang Yu and Zhongshi Pan

 The manuscript by Jinyan Gao et al. presents the framework for small IR target detection using a double linear array detector. The proposed mathematical modeling seems to be absolutely correct. The authors have made substantial contributions to this field of research. The importance of this study stems from the applicability of the developed platform to the extraction of the targets with a very low false alarm rate and an acceptable detection rate.

In my opinion, this paper can be recommended for publication.

There is only one question:

1. Are three experiments on scenes with different shapes of cloud enough to check the proposed small target detection methods?

Author Response

Thanks for your constructive comment. We have added another three experiments to enrich the performance of our proposed method, as shown in Fig. 10.

Reviewer 2 Report

The authors developed a mathematical model and experimentally demonstrated a new double linear array detector system for infrared small-dim target detection. This work is of interest to the detection research community where the infrared search and tracking (IRST) can capture targets from a long distance and has a strong anti-interference ability. The proposed method could solve a number of issues, such as, long target detection time, small image size, etc.

However, several issues need to be resolved before this manuscript could be published.

1)      On the page 1, line 2, the author described, “ the detector scans from the up to the bottom, image pairs are generated at short intervals. After…”. It is not clear what the author tried to imply by, “ up to the bottom”.

2)      “SCR” at page 1, line 6 was not defined previously.

3)      Annotations for Fig. 5 would be helpful for the readers.

4)      On page 7, the authors used the abbreviation “ROC” in the Fig. 6 caption, which was not explained before.

5)      From the model development and experimental section, it is not clear how this method is advantageous over other methods.

6)       Too few samples were analyzed for the experiment.

7)      In the conclusion, on the page 10, line 237, the authors, described, “ Monte Carlo experiments have been conducted on typical..”. However, in the experiment section, Monte Carlo method was not described.

Author Response

Comment 2.1

On the page 1, line 2, the author described, “ the detector scans from the up to the bottom, image pairs are generated at short intervals. After…”. It is not clear what the author tried to imply by, “ up to the bottom”.

Response 2.1

Thank you for the comment. To make the response clearer, we have redescribed the imaging process as follows.

“The special feature of the detector is that image pairs can be generated at short intervals in one scan.”(On the page 1, line 2)

Comment 2.2

SCR” at page 1, line 6 was not defined previously.

Response 2.2

Thank you for the suggestions. We have added the explanation for SCR as Signal-to-Clutter Ratio the first time it is mentioned (in the abstract, On the page 1, line 6).

Comment 2.3

Annotations for Fig. 5 would be helpful for the readers.

Response 2.3

Thank you for the suggestions. We have added the annotations for Fig. 5. (On the page 6)

Comment 2.4

On page 7, the authors used the abbreviation “ROC” in the Fig. 6 caption, which was not explained before.

Response

Thank you for the suggestions. We have added the explanation for ROC as Receiver Operating Characteristic on the page 7, line 181.

Comment 2.5

From the model development and experimental section, it is not clear how this method is advantageous over other methods.

Response 2.5

Thank you for the comment. We understand that it is better to add some comparison experiments to verify the effectiveness of our method. However, the task of this work is to detect infrared small targets imaged by the double linear array detector. Therefore, our work focuses on how to fully use spatial and temporal information in image pairs generated by the double linear array detector in one scan. We think that the model development and experimental section may not be optimal, but should be sufficient to draw a conclusion that the proposed method can be used to successfully detect targets with a low false alarm rate and an acceptable detection rate.

Comment 2.6

Too few samples were analyzed for the experiment.

Response 2.6

Thank you for the comment. We have added another three experiments to demonstrate the effectiveness of our method, as shown in Fig.10. (On the page 13)

Comment 2.7

In the conclusion, on the page 10, line 237, the authors, described, “ Monte Carlo experiments have been conducted on typical..”. However, in the experiment section, Monte Carlo method was not described.

Response 2.7

Thank you for the comment. Since the Receiver Operating Characteristic (ROC) curves are all obtained through Monte Carlo simulations, we have clarified this on the page 7, line 181 and on the page 9, line 207.

Reviewer 3 Report

In this paper authors present a complete framework for small infrared target detection using a double linear array detector. To begin, the simulation of a brand new double linear array detector that can produce image pairs at regular intervals is carried out. The image differentiation based change detection method in the temporal domain is proposed combined with the structure tensor edge suppression method in the spatial domain. This is done because of the limitations of singular spatial or temporal detection methods. On typical IR images, Monte Carlo simulations have been run, and the results of these simulations show that targets can be accurately extracted with a very low false alarm rate and an acceptable detection rate.

The article overall seems well written and worthy of publication. The references are also numerous and up-to-date, illustrating the state of the art in research.

Asking for more detail in the description of the methods and presentation of the results, I suggest a re-reading to eliminate inaccuracies in the formatting, very few. For example:

- Line 6: Signal-to-Clutter Ratio (SCR). Please, avoid acronyms in the abstract unless the acronym is used multiple times in the abstract. If an acronym is used in the abstract, it must be defined in the abstract, and then spelled out again the first time it is used in the body of the paper (as was correctly done in lines 176-177). So, add the explanation of the acronym the first time it is mentioned (in the abstract, line 6).

- Figures 1, 2, 3, 9, 10: Figures, as was correctly done for the others, should also be cited before placement in the main text.  

- For research articles with several authors, a short paragraph specifying their individual contributions must be provided, as required in the template, in "Author Contribution". 

Author Response

Comment 3.1

Line 6: Signal-to-Clutter Ratio (SCR). Please, avoid acronyms in the abstract unless the acronym is used multiple times in the abstract. If an acronym is used in the abstract, it must be defined in the abstract, and then spelled out again the first time it is used in the body of the paper (as was correctly done in lines 176-177). So, add the explanation of the acronym the first time it is mentioned (in the abstract, line 6).

Response 3.1

Thank you for the suggestions. We have added the explanation for SCR as Signal-to-Clutter Ratio the first time it is mentioned (in the abstract, line 6).

Comment 3.2

Figures 1, 2, 3, 9, 10: Figures, as was correctly done for the others, should also be cited before placement in the main text.

Response 3.2

Thank you for the suggestions. Figures 1, 2, 3, 9, 10 have been cited before placement in the main text.

Comment 3.3

For research articles with several authors, a short paragraph specifying their individual contributions must be provided, as required in the template, in "Author Contribution".

Response 3.3

Thank you for the comment. We have added the "Author Contribution" on the page 10, line 240-245.

Round 2

Reviewer 2 Report

Thanks for answering all the questions. 

1) Page 1, line 30 reads, " the singular spatial or temporal detection methods [? ][22][23]. They use features in both...".  Please fix the reference. 

2) Page 7, line 181 reads as, " The theoreticReceiver Operating Characteristic (ROC) curves validated through Monto". Please fix the spacing. 

Author Response

Thank you for the comment. Typos have been corrected in the revised manuscript.(On the Page 1, line 30 and on Page 7, line 181.)